# An open-source, lockable mouse wheel for the accessible implementation of time- and distance-limited elective exercise

**Joseph J. Bivona, III** [1,2] *, **Matthew E. Poynter** [1] *

**1** Department of Medicine and Vermont Lung Center, University of Vermont Larner College of Medicine, Burlington, Vermont, United States of America, **2** Cellular, Molecular, and Biomedical Sciences Doctoral Program, University of Vermont, Burlington, Vermont, United States of America

* JJ.Bivona@uvm.edu (JJB); matthew.poynter@med.uvm.edu (MEP)

**Data Availability Statement:** All model, code, and manual files are available for download at: https://github.com/jjbivona/LOSTwheel.

## Abstract

Current methods of small animal exercise involve either voluntary (wheel running) or forced (treadmill running) protocols. Although commonly used, each have several drawbacks which cause hesitancy to adopt these methods. While mice will instinctively run on a wheel, the distance and time spent running can vary widely. Forced exercise, while controllable, puts animals in stressful environments in which they are confined and often shocked for "encouragement." Additionally, both methods require expensive equipment and software, which limit these experiments to well-funded laboratories. To counter these issues, we developed a non-invasive mouse running device aimed to reduce handler-induced stress, provide time- and distance-based stopping conditions, and enable investigators with limited resources to easily produce and use the device. The Lockable Open-Source Training-Wheel (LOST-Wheel) was designed to be 3D printed on any standard entry-level printer and assembled using a few common tools for around 20 USD. It features an on-board screen and is capable of tracking distances, running time, and velocities of mice. The LOST-Wheel overcomes the largest drawback to voluntary exercise, which is the inability to control when and how long mice run, using a servo driven mechanism that locks and unlocks the running surface according to the protocol of the investigator. While the LOST-Wheel can be used without a computer connection, we designed an accompanying application to provide scientists with additional analyses. The LOST-Wheel Logger, an R-based application, displays milestones and plots on a user-friendly dashboard. Using the LOST-Wheel, we implemented a timed running experiment that showed distance-dependent decreases in serum myostatin as well as IL-6 gene upregulation in muscle. To make this device accessible, we are releasing the designs, application, and manual in an open-source format. The implementation of the LOST-Wheel and future iterations will improve upon existing murine exercise equipment and research.

**Funding:** This study was made possible by support from the Larner College of Medicine and the Vermont Lung Center T32HL076122. The funders had no role in study design, data collection and analysis, decision to publish, or preparation of the manuscript.

**Competing interests:** The authors have declared that no competing interests exist.

## Introduction

While exercise is a safe and effective health strategy [1], the ability for laboratories to test hypotheses concerning the physiological adaptations brought about by movement in mouse models is hindered by the expense of commercial products [2]. To understand the systemic effects of untrained exercise and lower the barrier of entry for murine-exercise research, we have developed an open-source mouse running wheel that accurately tracks and displays the distance traveled and restricts wheel running at specified distances or times. The Lockable Open-Source Training Wheel (LOST-Wheel) has been designed for both standalone and computer connected scenarios. The only requirement for standalone mode is a USB power source. In this mode, the LOST-Wheel can log cumulative distance, which is visualized through the onboard screen. When a computer is connected to the wheel through the LOST-Wheel Logger application, more detailed data such as speed and time running can be collected, graphed, and exported. The LOST-Wheel was tested in both overnight (acute) and week-long (chronic) experiments. Samples from the acute, untrained, bout of exercise were subjected to analysis with real-time quantitative polymerase chain reaction and protein quantification through Luminex assays.

This design of the LOST-Wheel was inspired as an attempt to create an inexpensive, freely accessible, and human-relevant model of exercise. One drawback of voluntary exercise is the inability to limit running distances [3]. For studies that interrogate the dose dependent effect of exercise or to restrict running to certain times and distances, we implemented a microcontroller regulated locking mechanism to prevent wheel movement at the will of the investigator. Previous work using commercially available products has shown that wheel running produces dose-dependent effects on neuron proliferation and dendritogenesis [4]. Additionally, that the presence of an immobile wheel in a cage also elicits neurological effects in the absence of its use demands that such a device-exposure group should be included as a proper control in wheel-running experiments [4, 5]. Despite the average gait of a mouse being 5–6 cm [6], mice voluntarily run upwards of 7 hours and 20 km/night [3] when provided with a standard wheel [7, 8]. The use of a wheel for long periods cannot be explained by a single theory [9], and physiological effects differ substantially between strains of mice [10]. By limiting running distances, the locking capacity of the LOST-Wheel enables researchers to normalize voluntary exercise across animals. While forced exercise (treadmill running) is an alternative strategy that ensures a consistent distance across animals, the handling [11], confinement [12], and electrical shock [13] required for its implementation can induce stress and alter the biological responses being studied [14].

To verify the effectiveness of the LOST-Wheel and to evaluate the consequences of a single, untrained bout of exercise, we allowed a cohort of mice to perform voluntary exercise for a single night (their waking time) then performed RT-qPCR on gastrocnemius muscle and multiplex analysis of several myokines in serum. We confirmed exercise-induced physiological changes, including increases in gastrocnemius *Il6* gene expression and a significant, negative, relationship between serum myostatin and the distance traveled by mice.

## Methods

### LOST-Wheel design and code

The LOST-Wheel was designed using Fusion360 (Autodesk, San Rafael, CA) and is composed of four pieces: main body, top face, servo pin, and wheel. Table 1 contains a component list for the electronics, bearings, axle, and hardware. Slicing the models for 3D printing was conducted using Cura (Ultimaker, Utrecht, Netherlands). All pieces were printed using fusion deposition

**Table 1. Component list for the LOST-Wheel.**

| Component | Quantity |
|---|---|
| 6 mm ID, 10 mm OD, x 3 mm bearing | 3 |
| 6 mm axle cut to 65 mm | 1 |
| M3x5 self-tapping screw | 1 |
| M2x6 self-tapping screw | 1 |
| M2.3x8 self-tapping screw | 8 |
| M1.7x6 self-tapping screw | 8 |
| 10 mm x 5 mm x 3 mm neodymium magnet | 2 |
| 22-gauge, 2.54 mm breadboard jumper wires, 3 male, 7 female | 10 |
| Arduino Nano (or similar) | 1 |
| 9g micro servo | 1 |
| KY-003 hall effect sensor | 1 |
| 0.96 inch 128x64 OLED Screen I2C connection SSD1306 Driver | 1 |

This list serves as a template for the electronics and hardware required for the device. Generic Arduino clones can be substituted as microcontrollers since they are often a fraction of the price. Magnet size and quantity can also be changed depending on availability and accuracy required.

modeling with polylactic acid (PLA) on an Ender3 V2 3D printer (Creality 3D, Shenzhen, China). The sketches used to program the LOST-Wheel were created in Arduino IDE (Arduino, New York City, NY) in Arduino/C++ language with the additional libraries, U8g2 and U8x8. A computer rendering, representative image, and wiring diagram for the LOST-Wheel are shown in Fig 1A–1C, respectively. In acute exercise experiments, the Timer Mode protocol was uploaded and set to begin when the wheel was powered on. For chronic exercise, the Distance Mode protocol was uploaded, and the threshold set to $10^6$ m for unlimited running. The LOST-Wheel can be powered indefinitely; however, in the experiments of this manuscript, data was collected daily, at which time wheels were reset.

All files required to build and program the LOST-Wheel are available at https://github.com/jjbivona/LOSTwheel. The LOST-Wheel design files, manual, and LOST-Wheel Logger software are licensed under a Creative Commons Attribution-NonCommercial-ShareAlike 4.0 International License. To access a build video and setup tutorial, visit: https://www.youtube.com/channel/UCUp9zD0H99VcX0XXl2qGUmg.

## LOST-Wheel logger

While the LOST-Wheel can accurately display distance using its onboard screen, more detailed information can be obtained using the LOST-Wheel Logger Application. The application was created in R and RStudio (version 4.1.1 and 1.4 respectively) using Shiny [15], ggplot2 [16], and Serial [17] packages. Users are instructed to enter an ID for the wheel, the associated communication port (COM port), and the duration of data collection. After information is entered and the start button is pressed, the logger restarts the wheel and collects data in one second intervals until the duration is met. The program then calculates the individual slopes between each data point to determine the maximum speed in meters/second. Using this information, the Logger can determine the amount of time the mouse has run. Previous literature indicates that untrained mice have an average speed of 1–2 km/h (0.28–0.56 m/s) [18]; therefore, a threshold of 0.2 m/s is applied to exclude non-running events. Finally, the LOST-Wheel Logger creates distance/time and velocity/time graphs and presents all information for the user. The number of wheels that can be simultaneously connected is limited by the number of

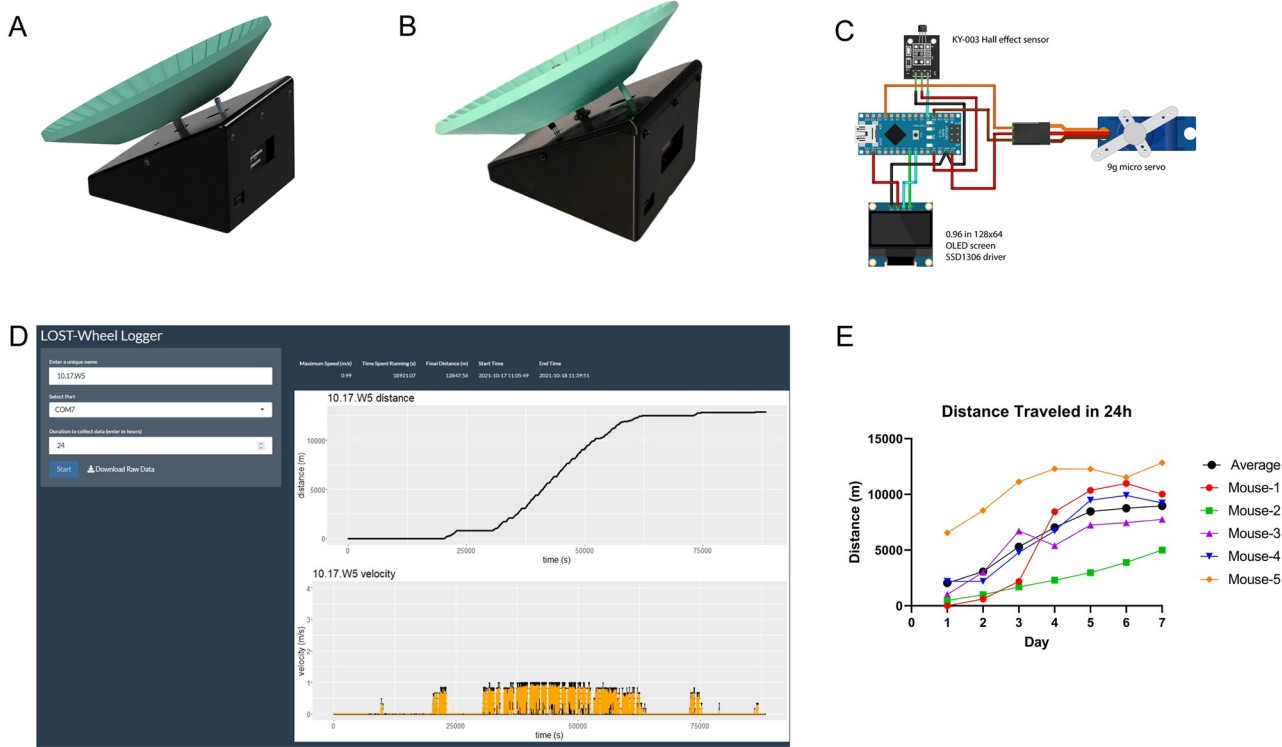

**Fig 1. Lost-Wheel assembly and testing.** A) Computer rendered design of the assembled LOST-Wheel. B) Completed assembly of the LOST-Wheel. C) Simplified wiring diagram of components. 22-gauge, 2.54 mm breadboard jumper wires are soldered to the microcontroller and connected to components using the attached plugs. Should a single component fail, this allows for easy replacement without resoldering. D) The LOST-Wheel Logger application can be used in conjunction with the LOST-Wheel to collect and plot additional data. E) Mice were allowed to run unrestricted for 7 days on the LOST-Wheel. Each morning, the distance was recorded, and wheels were reset (n = 5).

universal serial bus (USB) ports available on the computer. In our laboratory, we use an inexpensive USB hub to expand the number of wheels connected. The current version of the LOST-Wheel Logger accommodates a single wheel. Therefore, to collect data from multiple wheels, the user must open a separate instance of RStudio for each wheel.

## Mice

12–20 week old male C57BL/6J mice purchased from The Jackson Laboratory (Bar Harbor, ME) were housed in AAALAC-accredited animal facilities at the University of Vermont, and all experimental animal procedures were approved by the University of Vermont Institutional Animal Care and Use Committee, protocol #202100027. Mice were maintained on a 12 hour-light/dark cycle, beginning at 07:00 and 19:00, respectively, and provided chow and water *ad libitum*.

To examine the effect of a single, untrained bout of exercise, mice were brought from the vivarium and caged individually. A single LOST-Wheel, or an immobile "dummy" wheel, was introduced into each cage at 08:00. Wheels in the running group remained unlocked to acclimate the mice until 12:00, at which point the wheels locked. At 19:00 the wheels unlocked, and mice were allowed to run voluntarily for 12 hours, at which point they were euthanized by an intraperitoneal injection of pentobarbital (Euthasol, Midwest Veterinary Supply, Lakeville, MN), followed by exsanguination. Serum and gastrocnemius muscle were collected and snap frozen in liquid nitrogen.

For one week running experiments, wheels remained unlocked throughout the duration of the test. Distances were recorded at 09:00 each morning and the wheels were reset.

### RNA isolation, quantitative real-time polymerase chain reaction (qRT-PCR), and protein quantitation

Total RNA was isolated from liquid nitrogen pulverized whole muscle using TRIzol reagent followed by a chloroform-isopropanol extraction (Thermo Fisher Scientific, Waltham, MA, USA). RNA concentration and purity were measured using a NanoDrop 2000 spectrophotometer. (Thermo Fisher Scientific). cDNA was synthesized from 100 ng of RNA using the qScript Supermix reagent kit per manufacturer's instructions (Quantabio, Beverly, MA, USA). Quantitative real-time PCR was performed using iTaq Universal SYBR Green Supermix on a CFX96 Touch (Bio-Rad, Hercules, CA, USA), with the relative mRNA expression calculated using the threshold cycle (Ct; $2-\Delta\Delta Ct$) method normalized to *Gapdh* expression. The following primer sequences were used (Integrated DNA Technologies, Coralville, IA, USA): *Il6* forward `5'-CCCGGAGAGGAGACTTCACAG-3'`, reverse `5'-GAGCATTGGAAATTGGGGTA-3'`; *Gapdh* forward `5'-ACGACCCCTTCATTGACCTC-3'`, reverse `5'-TTCACACCCATCA CAAACAT-3'`.

Serum samples were analyzed using a Milliplex Mouse Myokine Magnetic Bead Panel (Millipore Sigma, St. Louis, MO, USA) on a Luminex 100 xMAP Instrument (Bio-Rad) according to kit instructions.

### Statistical analysis and figures

RT-qPCR and Milliplex data were analyzed and visualized using GraphPad Prism version 9.2.0 for Windows (GraphPad Software, San Diego, CA, USA) with unpaired t-tests and a linear regression, respectively. Significance is designated by p-values $< 0.05$. Fig 2A was created with BioRender.com.

## Results

### Building and testing the LOST-Wheel

All components in Table 1 are readily available and the 3D printed models can be created using any entry-level 3D printer capable of printing in polylactic acid (PLA) filament. The wheel can be assembled using only a #1 Phillips head screwdriver, a soldering iron, and a wire cutter/stripper. Excluding a 3D printer, the entire apparatus can be created for less than 15 USD. The models can easily be modified to account for larger diameter bearings and drive shafts, additional wheel magnets, or changes in component mounting holes. A representative rendering and completed wheel are shown in Fig 1A and 1B, respectively. A simplified component wiring diagram is shown in Fig 1C. We have also created a series of videos that outline the assembly, programming, and cage setup, which can be accessed at https://www.youtube. com/channel/UCUp9zD0H99VcX0XXl2qGUmg.

The LOST-Wheel was tested for 7 days to evaluate durability and animal safety, during which mice steadily increased distance traveled per day (Fig 1D), averaging 2046.38 ± 2647.11 m on the first day and 8972.71 ± 2888.14 m at the end of the trial. One mouse did not use the wheel until the second day.

### Locking criteria

To limit and control mouse exercise, we developed three separate modes for the LOST-Wheel. These are first edited by the user based on their experimental requirements and uploaded to

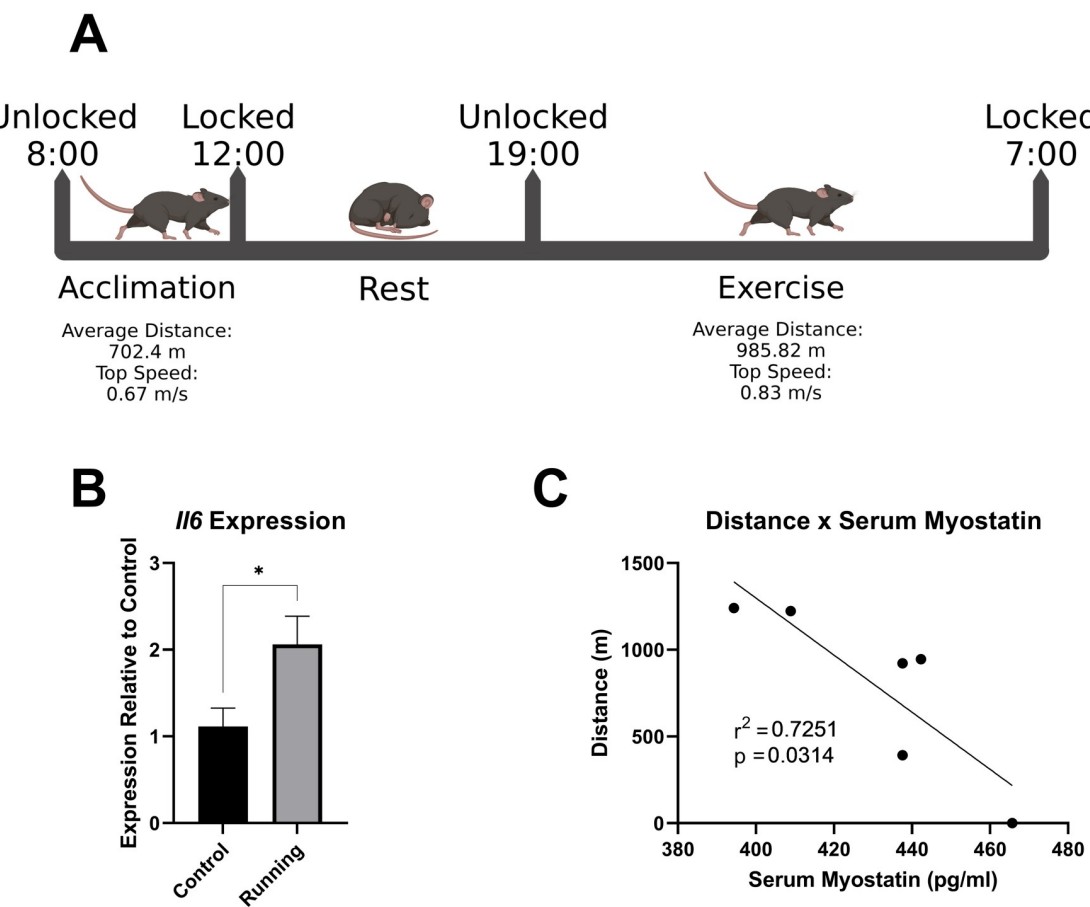

**Fig 2. Acute exercise of untrained mice.** A) Experimental setup. Mice were acclimated to the LOST-Wheel for four hours, at which point the wheel locked until the evening. Mice were allowed to run unrestricted for 12 hours, at which point the wheels relocked and mice were euthanized. B) RT-qPCR analysis of gastrocnemius muscle *Il6* expression (n = 6-7/group, unpaired t test). C) Serum myostatin concentrations relative to the distance traveled of mice undergoing untrained acute exercise (n = 6, simple linear regression).

the wheel through the Arduino IDE program. In Timer Mode, the user inputs the amount of time (in hours) for which they want the wheel to be sequentially locked and unlocked. This program was implemented for our acute voluntary exercise experiments. The second and third modes limit exercise by distance and time spent running, respectively, and allow for the normalization of exercise across animals within experimental groups.

## Acute voluntary exercise by untrained mice

To examine the effects of a single bout of voluntary exercise on untrained mice, we subjected mice to a one-day acclimation and exercise protocol (Fig 2A). Mice were singly placed in cages containing either a LOST-Wheel or an immobile "dummy" wheel at 08:00. The wheel remained unlocked for four hours for acclimation, at which point the servo inserted a pin into the wheel to lock the device (12:00). At 19:00, the pin was withdrawn, and mice were allowed to voluntarily use the wheel for 12 hours. At the end of exercise, we collected serum and leg muscles, then measured expression and production of interleukin-6 (IL-6), a muscle-produced cytokine (myokine) reported to be induced in exercised human subjects [19] as well as in

mouse models of acute exercise [20] and myostatin. We observed a significant increase in the expression of *Il6* in the gastrocnemius muscle following acute exercise (Fig 2B). Interestingly, there was a significant correlation between distance traveled on the LOST-Wheel and serum myostatin measured by Luminex (Fig 2C). One mouse did not log any distance on the LOST--Wheel. This mouse was excluded from RT-qPCR measurements but included in measurements of serum myostatin as a sedentary control.

## Discussion

This manuscript presents an open-source mouse wheel that can be affordably and efficiently assembled by investigators. The LOST-Wheel can be used in standalone mode to collect data on overall distances traveled. When used in combination with a computer, the wheel interfaces with the LOST-Wheel Logger application to calculate the amount of time spent running and top speed achieved. Finally, we verified the integrity of the device and observed physiological changes in serum and muscle gene expression from a single untrained bout of exercise.

### Physiological alterations after running

While the primary objective was to assess the efficacy of the LOST-Wheel, there was merit in observing the effects of acute exercise. Muscle derived signaling molecules, termed myokines, are released during muscle contraction and indicate physiological adaptations following exercise. Most notably, IL-6 is highly upregulated and is believed to function differently from classic inflammatory signaling by instead increasing glucose sensitivity and uptake [21]. We observed increases in *Il6* expression in the gastrocnemius muscle (Fig 2B); however, increased IL-6 protein concentrations were not detected in serum using a myokine multiplex panel, implicating its local effect in the muscle. Additionally, we observed a significant, negative, correlation between distance traveled and serum myostatin concentrations (Fig 2C). As a regulator of muscle growth and differentiation [22], this correlation implies that myostatin is dose dependently regulated by running distance. These results align with previous studies in humans and rats, in which myostatin was transiently decreased after bouts of acute exercise [23, 24].

### Voluntary exercise versus forced exercise

While treadmill based forced exercise allows for controlled speed, duration, and incline of training, it increases corticosterone and norepinephrine levels, indicating a strong stress response that is not elicited by during voluntary exercise [25–28]. The shock, confinement, or handling of the mice can all contribute to the increased stress reported. Additionally, the presence of an immobile wheel can have the added benefit of environmental enrichment [29, 30]. Due to this effect, we advise using a locked LOST-Wheel or creating a dummy wheel (fully assembled without electronic components) for control groups [4, 5].

### Durability of the LOST-Wheel

Several iterations of the LOST-Wheel were prototyped before using the Hall effect sensor and magnet combination. Originally, the wheel rotated on a rotary encoder, an electro-mechanical part that has a finite number of rotations (30,000–100,000) before wearing out. We also attempted using an infrared sensor, but cage bedding would often block the beam, rendering it useless. The magnet and Hall effect sensor bypass these problems and should remain operational indefinitely. The running surface and shaft can easily be removed and sprayed with ethanol to disinfect between uses. While mice have occasionally chewed the running surface, this

has not hindered its balance or performance. We have also designed and provided a guard for the power cord that protects it from destructive animals.

## Comparison to existing wheels

To verify data collection on the LOST-Wheel, we have compared our 7-day protocol to the well detailed and extensive work of de Bono and colleagues [18], who developed a method to track distances run by mice but that lacks the ability to limit exercise, requires an expensive data acquisition unit with software (Spike 2, https://ced.co.uk/prices/1401options, https://ced.co.uk/prices/softwareprices), and has limited customizability and programmability. We have developed the LOST-Wheel Logger App, which allows for detailed collection, analysis, and export of data similar to the Spike 2 program. Reassuringly, our device recapitulates the distances reported in the aforementioned work. Recently, an open-source wheel was introduced that tracks running distances using a similar Hall Effect sensor as the LOST-Wheel. However, this device does not allow for controlled exercise or additional data collection as it only collects cumulative distance measurements [31]. Forced exercise designs exist that can be used to exercise mice, but such devices require extensive machining and calibration [32] or the repurposing of an existing human-treadmill [33].

## Cost

The LOST-Wheel provides an inexpensive alternative to commercial murine exercise devices. Similar distance tracking devices, without locking abilities, cost 300–400 USD per wheel, require an additional data acquisition unit (400–900 USD), and necessitate accompanying software (800–2500 USD) (price quotes are from correspondence with commercial retailers). The availability and shallow learning curve of Arduino-based microcontrollers allows for laboratories to create their own devices at a fraction of the price [34]. The 3D printed parts can be outsourced to university fabrication labs, commercial 3D print operations, or fabricated in-house as entry level fused deposition modeling printers have substantially decreased in price over the last decade, with entry level printers ranging from 200–300 USD.

## Future designs

The LOST-Wheel was created out of necessity and to improve current research-based exercise protocols in experimental animals. We have shown that this inexpensive, open-source wheel can provide investigator-controllable exercise to small rodent research without modification of the cage. The value of an open-source project is that it allows researchers to easily implement changes that fit their research goals and budgets. Future iterations of the LOST-Wheel and Logger App can include wireless transmission of data to a smartphone through commercially available Bluetooth-Arduino adaptors. Other changes may include on-board data storage, operant conditioning modifications, or using the locking pin to provide resistance against the wheel to model weighted wheel hypertrophy-inducing exercise [35].

## Supporting information

**S1 Data. The code, manual, and 3D files in both .STL and .F3D format, can be found at: www.github.com/jjbivona/lostwheel.**
(TXT)

**S1 Video. Videos for building and setting up the LOST-Wheel can be found at: https://www.youtube.com/channel/UCUp9zD0H99VcX0XXl2qGUmg.**
(TXT)

## Acknowledgments

The authors thank Jeffery L. Brabec and Colleen E. Yancey for their help debugging code.

## Author Contributions

**Conceptualization:** Joseph J. Bivona, III.

**Data curation:** Joseph J. Bivona, III.

**Formal analysis:** Joseph J. Bivona, III.

**Funding acquisition:** Joseph J. Bivona, III.

**Investigation:** Joseph J. Bivona, III.

**Methodology:** Joseph J. Bivona, III.

**Project administration:** Joseph J. Bivona, III, Matthew E. Poynter.

**Software:** Joseph J. Bivona, III.

**Supervision:** Matthew E. Poynter.

**Visualization:** Joseph J. Bivona, III.

**Writing – original draft:** Joseph J. Bivona, III.

**Writing – review & editing:** Joseph J. Bivona, III, Matthew E. Poynter.

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
