## [Decision Letter · Decision Letter 0]

22 Nov 2021

PONE-D-21-33956An open source, lockable mouse wheel for the accessible implementation of time- and distance-limited elective exercisePLOS ONE

Dear Dr. Bivona III,

Thank you for submitting your manuscript to PLOS ONE. After careful consideration, all the reviewers were enthusiastic about the manuscript but there are a few minor changes hat will be necessary to finalize the manuscript.  These are fairly minor and can be re-reviewed editorially without need to resend to reviewers.  . Therefore, we invite you to submit a revised version of the manuscript that addresses the points raised during the review process. ==============================

As you can see in the reviews, there are just a few changes required.  These mostly involve addition of references as well as the notation of two of the reviewers that you should eliminate or reword the use of the word hyperphsyiological regarding the amount of exercise performed by mice (they seen to run approximately 3-4km/night  under voluntary conditions (see Gerecke et al, 2010) and others referenced by reviewers.   Additionally, reviewer 3 asks for some more information on durability of the wheels as well as about how many can be run on a single computer (or can this be expanded by addition of a COM port hub? I look forward to your revised manuscript which should be of benefit to the scientific community.

We look forward to receiving your revised manuscript.

Kind regards,

Richard Jay Smeyne

Academic Editor

PLOS ONE

Journal Requirements:

Reviewers' comments:

Reviewer's Responses to Questions

**Comments to the Author**

1. Is the manuscript technically sound, and do the data support the conclusions?

Reviewer #1: Yes

Reviewer #2: Yes

Reviewer #3: Yes

2. Has the statistical analysis been performed appropriately and rigorously? 

Reviewer #1: Yes

Reviewer #2: Yes

Reviewer #3: N/A

3. Have the authors made all data underlying the findings in their manuscript fully available?

Reviewer #1: Yes

Reviewer #2: Yes

Reviewer #3: Yes

4. Is the manuscript presented in an intelligible fashion and written in standard English?

Reviewer #1: Yes

Reviewer #2: Yes

Reviewer #3: Yes

5. Review Comments to the Author

Reviewer #1: This communication provides an open source-based device as a means to study physical exercise, as modelled by wheel-running, in mice. The main issues which are successfully dealt with are both scientific (a device which permits to fix time and/or distance limits according to the experimenter will) and practical (a minor cost, compared to commercially available devices). I encourage the reader to long into the link providing the (detailed) video related to piece assembly.

My sole critics concern bibliographical issues, which however need to be taken into account:

(1) In the 2nd par. of the Introduction, it is indicated that "one drawback of voluntary exercise is the inability to limit running distances". In the authors' mind such a pitfall might probably extend to running times/periods. First, I encourage the authors to read two studies (Dostes et al., Hippocampus 2016; Dubreucq et al., Biol. Psychiatry 2013) wherein the time allowed to run was varied (showing e.g. that exercise-induced neurogenesis is running duration-dependent). Second, whether it is more meaningful to limit running distance or running duration is an open question. Indeed, rodents run for hours but these consist in successive, albeit extremely short, episodes with resting pauses aimed e.g. at drinking, resting, exploring or eating. Moreover, mice do run during the dark phase of the light/dark cycle but their maximal effort is observed during the first 3-4 hours (see the above mentioned study by Dubreucq as an illustration). Taken with the observation that each mouse bears its own active period and its own running speed, it is maybe more relevant to limit the time, rather than the distance, as the variable for wheel locking/unlocking.

(2) In the same paragraph, the authors use the word "hyperphysiological" for mouse running behaviour. I sincerely believe that this word should be removed because it might be prone to anthropomorphism. As humans, we consider these distances as enormous. However, mice (and rats) are able to do so without constraint. I encourage the authors to read the excellent review of Sherwin (Animal Behav 1998), which in addition, will allow the authors to change several of their (recent) references by older one which might be more relevant.

(3) Legends to the two figures should be separated from the main text and place at the end before the figures

(4) Discussion:

• The fact that circulating BDNF is not increased by acute exercise is somewhat surprising. How do the authors explain this observation?

• The paragraph comparing forced vs voluntary exercise is incorrect. Asz it stands, it seems that voluntary exercise does not increase corticosterone levels. Besides the observation that it is increased with wheel-running (see relevant figures in ref 22), the fact that corticosterone increases with exercise is logical and is also observed in humans (changing corticosterone to cortisol) because it provides a source of energy.

• The authors should mention the fact that their device is considered an environmental enrichment which, per se, can alter physiological and behavioural variables (see the abundant bibliography on that subject) hence indicating the need to host control mice with locked wheels (see the abovementioned study by Dostes et al. as an additional illustration).

• As mentioned by the authors, "free" wheel-running devices as the one reported in this study allow to easilly measure running performance under voluntary dimensions. However, these devices, by their "free access" dimension, do not permit to selectively measure running motivation. The latter requires operant conditioning paradigms (and hence expensive setups) wherein running access is rendered possible by prior lever pressing (see Iversen, J Exp Anal Behav 1993) or nose poking (see Muguruza et al., JCI Insight 2019) under fixed or variable ratio reinforcement schedules.

Reviewer #2: The article, regarding the building and validity of the LOSTwheel is very well done. The authors present clear information about materials, assembly, and implementation of the equipment. In addition, they do appropriate tests to verify their observational analysis of the running behavior of the mice. This is true both for not only whether they run or not, as well as for the distance that they ran. Thus, these results are in keeping with many other studies regarding wheel running behavior in mice, indicating that the design of the LOSTwheel is valid for running behavior. Importantly, access to a low cost strategy to make these wheels is invaluable to the many scientists who study this, but do not have large grants to buy the expensive monitoring cages available on the market. Thus, this research has broad implications for impacting the field, including to make this research more widely available. As someone with little experience with electrical builds, I appreciate the very comprehensive manual with clear, step by step pictures and instructions, as well as the online tutorials. The level of detail and clarity provided make this easily replicable.

Reviewer #3: This open source, lockable mouse wheel is a useful tool that may enable investigators to conduct mouse exercise trials without limitations due to equipment cost.

Some clarification in the manuscript statements is necessary:

1. Voluntary running wheels that allow restriction of running in time or distance increments are available, although the cost of the equipment may be prohibitive for some laboratories. This is not clear in all sections of the manuscript.

2. Previously published articles, referenced in the current manuscript (see De Bono et al., 2006) and (Gerecke et al., 2010) illustrating the amount of voluntary running that mice will perform in a given period of time (1 day to 90 days) refute the suggestion that mice will voluntarily run at ‘hyperphysiological” levels.

3. Some details about the potential protocols for the LOST-Wheel would be useful if clearly stated in the manuscript. For example, is one week (chronic), the maximum period of time for an exercise protocol (without restarting)?

How many wheels and COM ports may be run in a single experiment?

Approximately how long do the 3D printed wheel and magnet apparatus work efficiently with continued use and cleaning?

The Lost-Wheel manual is clear and appears easy to follow to build the wheel and controlling components. The video links provided are very helpful, however, parts of the Assembly video, especially the final assembly portion of the box housing the microcontroller, Nano, and Hall effect sensor, as well as the connection of the components would benefit from a closer view.

Minor comments:

The use of Gapdh for normalization of the expression of the gene of interest in qPCR is not a stable comparator in all cases (see(Desseille et al., 2017; Xu et al., 2018; Hildyard et al., 2019)). The housekeeping gene 18S rRNA has invariant expression and provides more stable comparison.

Reference #13 should be DeBono JPD (De Bono et al., 2006).

References:

De Bono JP, Adlam D, Paterson DJ, Channon KM (2006) Novel quantitative phenotypes of exercise training in mouse models. American journal of physiology Regulatory, integrative and comparative physiology 290:R926-934.

Desseille C, Deforges S, Biondi O, Houdebine L, D'Amico D, Lamazière A, Caradeuc C, Bertho G, Bruneteau G, Weill L, Bastin J, Djouadi F, Salachas F, Lopes P, Chanoine C, Massaad C, Charbonnier F (2017) Specific Physical Exercise Improves Energetic Metabolism in the Skeletal Muscle of Amyotrophic-Lateral- Sclerosis Mice. Front Mol Neurosci 10:332.

Gerecke KM, Jiao Y, Pani A, Pagala V, Smeyne RJ (2010) Exercise protects against MPTP-induced neurotoxicity in mice. Brain Res 1341:72-83.

Hildyard JCW, Finch AM, Wells DJ (2019) Identification of qPCR reference genes suitable for normalizing gene expression in the mdx mouse model of Duchenne muscular dystrophy. PLoS One 14:e0211384.

Xu H, Ren X, Lamb GD, Murphy RM (2018) Physiological and biochemical characteristics of skeletal muscles in sedentary and active rats. J Muscle Res Cell Motil 39:1-16.

6. PLOS authors have the option to publish the peer review history of their article (what does this mean?). If published, this will include your full peer review and any attached files.

Reviewer #1: **Yes: **Chaouloff

Reviewer #2: No

Reviewer #3: No

---

## [Author Response · Author response to Decision Letter 0]

26 Nov 2021

Dear Dr. Smeyne,

We are excited to resubmit our manuscript, “An open-source, lockable mouse wheel for the accessible implementation of time- and distance-limited elective exercise” to PLOS One. The comments we received were helpful to complete the revision and are addressed below. 

Best Regards,

Joseph J. Bivona III

Matthew E. Poynter

 

Reviewer #1: 

This communication provides an open source-based device as a means to study physical exercise, as modelled by wheel-running, in mice. The main issues which are successfully dealt with are both scientific (a device which permits to fix time and/or distance limits according to the experimenter will) and practical (a minor cost, compared to commercially available devices). I encourage the reader to long into the link providing the (detailed) video related to piece assembly.

My sole critics concern bibliographical issues, which however need to be taken into account:

(1) In the 2nd par. of the Introduction, it is indicated that "one drawback of voluntary exercise is the inability to limit running distances". In the authors' mind such a pitfall might probably extend to running times/periods. First, I encourage the authors to read two studies (Dostes et al., Hippocampus 2016; Dubreucq et al., Biol. Psychiatry 2013) wherein the time allowed to run was varied (showing e.g. that exercise-induced neurogenesis is running duration-dependent). Second, whether it is more meaningful to limit running distance or running duration is an open question. Indeed, rodents run for hours but these consist in successive, albeit extremely short, episodes with resting pauses aimed e.g. at drinking, resting, exploring or eating. Moreover, mice do run during the dark phase of the light/dark cycle but their maximal effort is observed during the first 3-4 hours (see the above mentioned study by Dubreucq as an illustration). Taken with the observation that each mouse bears its own active period and its own running speed, it is maybe more relevant to limit the time, rather than the distance, as the variable for wheel locking/unlocking.

We would like to thank Reviewer #1 for their kind words and constructive response. Our revised submission has included the following references in the second paragraph of the introduction and within the discussion as they further provide the case for appropriate controls and dose dependent effects of exercise:

Dostes S, Dubreucq S, Ladevèze E, Marsicano G, Abrous DN, Chaouloff F, et al. Running per se stimulates the dendritic arbor of newborn dentate granule cells in mouse hippocampus in a duration-dependent manner. Hippocampus. 2016;26(3):282-8. doi: https://doi.org/10.1002/hipo.22551.

Dubreucq S, Marsicano G, Chaouloff F. Emotional consequences of wheel running in mice: Which is the appropriate control? Hippocampus. 2011;21(3):239-42. doi: https://doi.org/10.1002/hipo.20778.

(2) In the same paragraph, the authors use the word "hyperphysiological" for mouse running behaviour. I sincerely believe that this word should be removed because it might be prone to anthropomorphism. As humans, we consider these distances as enormous. However, mice (and rats) are able to do so without constraint. I encourage the authors to read the excellent review of Sherwin (Animal Behav 1998), which in addition, will allow the authors to change several of their (recent) references by older one which might be more relevant.

Both Reviewer #1 and #3 have raised concerns about the use of hyperphysiological and upon examination, we agree that this term should be removed from the manuscript. Consequently, we have done so. We have also added the recommended paper (Sherwin, 1998) to the introduction as we enjoyed this perspective on the reasons why animals utilize running wheels. With our background in immunology, we often do not appreciate the behavioral aspects of murine research that the review highlights. 

(3) Legends to the two figures should be separated from the main text and place at the end before the figures

We have reviewed the PLOS One guidelines and believe the format is correct. 

(4) Discussion:

• The fact that circulating BDNF is not increased by acute exercise is somewhat surprising. How do the authors explain this observation?

In our experiment, BDNF was not detectable in the serum of both the control and exercise groups. We have removed this sentence from the results and discussion since BDNF and other myokines are not included as data and do not contribute to the manuscript. In humans, BDNF appears to be transiently found after exercise and is returned to baseline after about 10 minutes (Schmidt-Kassow et al 2012). We speculate this rapid utilization may also be the case in mice. It would appear other groups have also had trouble detecting serum BDNF (Delezie et al., 2019, Klein et al., 2010). 

Schmidt-Kassow M, Schädle S, Otterbein S, Thiel C, Doehring A, Lötsch J, Kaiser J. Kinetics of serum brain-derived neurotrophic factor following low-intensity versus high-intensity exercise in men and women. Neuroreport. 2012 Oct 24;23(15):889-93. doi: 10.1097/WNR.0b013e32835946ca. PMID: 22964969.

BDNF is a mediator of glycolytic fiber-type specification in mouse skeletal muscle

Julien Delezie, Martin Weihrauch, Geraldine Maier, Rocío Tejero, Daniel J. Ham, Jonathan F. Gill, Bettina Karrer-Cardel, Markus A. Rüegg, Lucía Tabares, Christoph Handschin. Proceedings of the National Academy of Sciences Aug 2019, 116 (32) 16111-16120; DOI: 10.1073/pnas.1900544116 

Klein AB, Williamson R, Santini MA, Clemmensen C, Ettrup A, Rios M, Knudsen GM, Aznar S. Blood BDNF concentrations reflect brain-tissue BDNF levels across species. Int J Neuropsychopharmacol. 2011 Apr;14(3):347-53. doi: 10.1017/S1461145710000738. Epub 2010 Jul 7. PMID: 20604989.

• The paragraph comparing forced vs voluntary exercise is incorrect. Asz it stands, it seems that voluntary exercise does not increase corticosterone levels. Besides the observation that it is increased with wheel-running (see relevant figures in ref 22), the fact that corticosterone increases with exercise is logical and is also observed in humans (changing corticosterone to cortisol) because it provides a source of energy.

Upon reviewing literature, we have included several sources that support the statement that voluntary exercise stimulates reduced levels of corticosterone compared to forced exercise. At Reviewer #1’s suggestion, we have removed the statement that voluntary exercise increases corticosterone levels and replaced it with mention that the wheel itself is an enrichment device. 

• The authors should mention the fact that their device is considered an environmental enrichment which, per se, can alter physiological and behavioural variables (see the abundant bibliography on that subject) hence indicating the need to host control mice with locked wheels (see the abovementioned study by Dostes et al. as an additional illustration).

We appreciate this suggestion and have added it into the discussion.

• As mentioned by the authors, "free" wheel-running devices as the one reported in this study allow to easilly measure running performance under voluntary dimensions. However, these devices, by their "free access" dimension, do not permit to selectively measure running motivation. The latter requires operant conditioning paradigms (and hence expensive setups) wherein running access is rendered possible by prior lever pressing (see Iversen, J Exp Anal Behav 1993) or nose poking (see Muguruza et al., JCI Insight 2019) under fixed or variable ratio reinforcement schedules.

The value of an open-source design is that all files can be modified (and we encourage it). We are in the early stages of creating a resisted LOST-Wheel for hypertrophic exercise. Based on the review by Sherwin and the references suggested by the reviewer, there is certainly a need to incorporate an operant condition feature. We would be willing to collaborate and help design such a device. 

Reviewer #2: 

The article, regarding the building and validity of the LOSTwheel is very well done. The authors present clear information about materials, assembly, and implementation of the equipment. In addition, they do appropriate tests to verify their observational analysis of the running behavior of the mice. This is true both for not only whether they run or not, as well as for the distance that they ran. Thus, these results are in keeping with many other studies regarding wheel running behavior in mice, indicating that the design of the LOSTwheel is valid for running behavior. Importantly, access to a low cost strategy to make these wheels is invaluable to the many scientists who study this, but do not have large grants to buy the expensive monitoring cages available on the market. Thus, this research has broad implications for impacting the field, including to make this research more widely available. As someone with little experience with electrical builds, I appreciate the very comprehensive manual with clear, step by step pictures and instructions, as well as the online tutorials. The level of detail and clarity provided make this easily replicable.

We thank Reviewer 2 for their time to review our manuscript and their encouraging comments. One of our goals was to make the assembly and coding of the LOST-Wheel as approachable as possible so that others can implement it in their labs. We hope that this is the case. 

Reviewer #3: 

This open source, lockable mouse wheel is a useful tool that may enable investigators to conduct mouse exercise trials without limitations due to equipment cost.

Some clarification in the manuscript statements is necessary:

1. Voluntary running wheels that allow restriction of running in time or distance increments are available, although the cost of the equipment may be prohibitive for some laboratories. This is not clear in all sections of the manuscript.

We have edited a sentence in the introduction to address this point and maintain this perspective through the remainder of the manuscript. 

2. Previously published articles, referenced in the current manuscript (see De Bono et al., 2006) and (Gerecke et al., 2010) illustrating the amount of voluntary running that mice will perform in a given period of time (1 day to 90 days) refute the suggestion that mice will voluntarily run at ‘hyperphysiological” levels.

We have been convinced through the comments of Reviewer #3 and #1 that the use of hyperphysiological is incorrect and anthropomorphizes the typical distances that mice run. Thus, we have removed these sentences from the manuscript.

3. Some details about the potential protocols for the LOST-Wheel would be useful if clearly stated in the manuscript. For example, is one week (chronic), the maximum period of time for an exercise protocol (without restarting)?

We have included additional information into the methods section that indicate the modes that were uploaded. 

How many wheels and COM ports may be run in a single experiment?

We have added a sentence in the methods to address this question. The number of wheels that can be run is limited by the number of USB ports available on the computer. In our laboratory, we use an inexpensive USB hub splitter to expand the number of wheels connected. The current version of the LOST-Wheel Logger can only accommodate a single wheel. Therefore, to collect data from multiple wheels, the user must open a separate instance of RStudio for each wheel. This is a minor inconvenience but works effectively. 

Approximately how long do the 3D printed wheel and magnet apparatus work efficiently with continued use and cleaning?

We have added the following paragraph to the discussion to address this comment:

Several iterations of the LOST-Wheel were prototyped before using the Hall effect sensor and magnet combination. Originally, the wheel rotated on a rotary encoder, an electro-mechanical part that has a finite number of rotations (30,000-100,000) before wearing out. We also attempted using an infrared sensor, but cage bedding would often block the beam, rendering it useless. The magnet and Hall effect sensor bypass these problems and should remain operational indefinitely. The running surface and shaft can easily be removed and sprayed with ethanol to disinfect between uses. While mice have occasionally chewed the running surface, this has not hindered its balance or performance. We have also designed and provided a guard for the power cord that protects it from destructive animals.

The Lost-Wheel manual is clear and appears easy to follow to build the wheel and controlling components. The video links provided are very helpful, however, parts of the Assembly video, especially the final assembly portion of the box housing the microcontroller, Nano, and Hall effect sensor, as well as the connection of the components would benefit from a closer view.

We appreciate the reviewer’s comments on the assembly video and plan to increase the size of the images projected in the sections pertaining to the final assembly during future updates.

Minor comments:

The use of Gapdh for normalization of the expression of the gene of interest in qPCR is not a stable comparator in all cases (see(Desseille et al., 2017; Xu et al., 2018; Hildyard et al., 2019)). The housekeeping gene 18S rRNA has invariant expression and provides more stable comparison.

We acknowledge that housekeeping genes are often the topic of debate. Yet, determining the best one may be situation specific. It appears through the references that for dystrophic, atrophic, and recovering muscle, 18S is a proven candidate. For exercise, it may be worth investigating 18S further, since ribosomal biogenesis can be linked to exercise induced hypertrophy. 

Wang X, Zhao H, Ni J, Pan J, Hua H, Wang Y. Identification of suitable reference genes for gene expression studies in rat skeletal muscle following sciatic nerve crush injury. Mol Med Rep. 2019 May;19(5):4377-4387. doi: 10.3892/mmr.2019.10102. Epub 2019 Mar 28. PMID: 30942461; PMCID: PMC6472138.

Regulation of Ribosome Biogenesis in Skeletal Muscle Hypertrophy. Vandré Casagrande Figueiredo and John J. McCarthy. Physiology 2019 34:1, 30-42 

Journal Requirements:

We have modified our revised manuscript to comply with PLOS ONE’s style requirements.

2. We note that the grant information you provided in the ‘Funding Information’ and ‘Financial Disclosure’ sections do not match. When you resubmit, please ensure that you provide the correct grant numbers for the awards you received for your study in the ‘Funding Information’ section

We have confirmed our grant numbers on the manuscript to match those on the Editorial Manager portal. 

We have determined that the data omitted do not add value to the manuscript and have removed the sentences. 

We have updated our methods to include the UVM IACUC protocol number.

We have searched our references with the retraction watch database and found no overlap.

---

## [Editor Report · Decision Letter 1]

7 Dec 2021

An open-source, lockable mouse wheel for the accessible implementation of time- and distance-limited elective exercise

PONE-D-21-33956R1

Dear Dr. Bivona III,

We’re pleased to inform you that your manuscript has been judged scientifically suitable for publication and will be formally accepted for publication once it meets all outstanding technical requirements.

Kind regards,

Richard Jay Smeyne

Academic Editor

PLOS ONE

Additional Editor Comments (optional):

I am excited for this paper to be published in PLoS One and feel that this may have a significant impact on the field; due to its ease of use and low cost. On a personal level, having worked in exercise, I look forward to its publication, at which point I will access the information and give this a try. Thank you for an nice piece of work.
---

## [Editor Report · Acceptance letter]

10 Dec 2021

PONE-D-21-33956R1 

An open-source, lockable mouse wheel for the accessible implementation of time- and distance-limited elective exercise 

Dear Dr. Bivona III:

I'm pleased to inform you that your manuscript has been deemed suitable for publication in PLOS ONE. Congratulations! Your manuscript is now with our production department. 

Kind regards, 

on behalf of

Dr. Richard Jay Smeyne 

Academic Editor

PLOS ONE